# Evaluation of Ibuprofen Use on the Immune System Indicators and Force in Disabled Paralympic Powerlifters of Different Sport Levels

**DOI:** 10.3390/healthcare10071331

**Published:** 2022-07-18

**Authors:** Felipe J. Aidar, Guacira S. Fraga, Márcio Getirana-Mota, Anderson Carlos Marçal, Jymmys L. Santos, Raphael Fabricio de Souza, Lucio Marques Vieira-Souza, Alexandre Reis Pires Ferreira, Dihogo Gama de Matos, Paulo Francisco de Almeida-Neto, Nuno Domingos Garrido, Alfonso López Díaz-de-Durana, Beat Knechtle, Breno Guilherme de Araújo Tinoco Cabral, Eugenia Murawska-Ciałowicz, Hadi Nobari, Ana Filipa Silva, Filipe Manuel Clemente, Georgian Badicu

**Affiliations:** 1Graduate Program of Physical Education, Federal University of Sergipe (UFS), São Cristovão 49100-000, Brazil; fjaidar@gmail.com (F.J.A.); guacirafraga@yahoo.com.br (G.S.F.); marcio_getirana@hotmail.com (M.G.-M.); acmarcal@yahoo.com.br (A.C.M.); jymmys.lopes@gmail.com (J.L.S.); raphaelctba20@hotmail.com (R.F.d.S.); profedf.luciomarkes@gmail.com (L.M.V.-S.); 2Group of Studies and Research of Performance, Sport, Health and Paralympic Sports (GEPEPS), Federal University of Sergipe (UFS), São Cristovão 49100-000, Brazil; 3Department of Physical Education, Federal University of Sergipe (UFS), São Cristovão 49100-000, Brazil; 4Graduate Program of Physiological Science, Federal University of Sergipe (UFS), São Cristovão 49100-000, Brazil; 5Department of Physical Education, State Univerity of Minas Gerais (UEMG), Passos 37900-106, Brazil; 6College of Physical Education and Exercise Science, University of Brasília (UnB), Brasília 70910-900, Brazil; alexandreispf@gmail.com; 7Cardiovascular & Physiology of Exercise Laboratory, University of Manitoba, Winnipeg, MB R3T 2N2, Canada; dihogogmc@hotmail.com; 8Department of Physical Education, Federal University of Rio Grande do Norte, Natal 59064-741, Brazil; paulo220911@hotmail.com (P.F.d.A.-N.); brenotcabral@gmail.com (B.G.d.A.T.C.); 9Research Center in Sports Sciences, Health Sciences and Human Development (CIDESD), University of Trás-os-Montes e Alto Douro, 5001-801 Vila Real, Portugal; ngarrido@utad.pt; 10Sports Department, Physical Activity and Sports Faculty-INEF, Universidad Politécnica de Madrid, 28040 Madrid, Spain; alfonso.lopez@upm.es; 11Institute of Primary Care, University of Zurich, 8091 Zurich, Switzerland; beat.knechtle@hispeed.ch; 12Medbase St. Gallen Am Vadianplatz, 9001 St. Gallen, Switzerland; 13Physiology and Biochemistry Department, Wroclaw University of Health and Sport Sciences, 51-612 Wroclaw, Poland; eugenia.murawska-cialowicz@awf.wroc.pl; 14Department of Exercise Physiology, Faculty of Educational Sciences and Psychology, University of Mohaghegh Ardabili, Ardabil 56199-11367, Iran; hadi.nobari1@gmail.com; 15Faculty of Sport Sciences, University of Extremadura, 10003 Cáceres, Spain; 16Escola Superior de Desporto e Lazer, Instituto Politécnico de Viana do Castelo, Rua Escola Industrial e Comercial de Nun’Álvares, 4900-347 Viana do Castelo, Portugal; anafilsilva@gmail.com (A.F.S.); filipe.clemente5@gmail.com (F.M.C.); 17Research Center in Sports Sciences, Health Sciences and Human Development (CIDESD), Polytechnic Institute of Maia, Maia, 5001-801 Vila Real, Portugal; 18Instituto de Telecomunicações, Delegação da Covilhã, 1049-001 Lisboa, Portugal; 19Department of Physical Education and Special Motricity, Transilvania University of Brasov, 500068 Brasov, Romania

**Keywords:** muscle strength, ibuprofen, paralympic powerlifting, immune system, recovery of function

## Abstract

Background: Paralympic powerlifting (PP) training is typically intense and causes fatigue and alterations in the immune system. Objective: To analyze whether IBU would affect performance and the immune system after training in PP. Methodology: 10 athletes at the national level (NL) and 10 at the regional level (RL) participated in the study, where force and blood indicators were evaluated after training. The study took place over three weeks: (1) familiarization and (2 and 3) comparison between recovery methods, with ibuprofen or placebo (IBU vs. PLA), 800 mg. In the evaluation of the force, the peak torque (PT), fatigue index (FI), and blood immune system biomarkers were analyzed. The training consisted of five sets of five repetitions with 80% of one maximum repetition (5 × 5, 80% 1RM) on the bench press. Results: The PT at the national level using IBU was higher than with PLA (*p* = 0.007, η2p = 0.347), and the FI in the NL was lower with IBU than with PLA (*p* = 0.002, η2p = 0.635), and when comparing the use of IBU, the NL showed less fatigue than the regional level (*p* = 0.004, η2p = 0.414). Leukocytes, with the use of IBU in the NL group, were greater than in the RL (*p* = 0.001, η2p = 0.329). Neutrophils, in the NL with IBU, were greater than in the RL with IBU and PLA (*p* = 0.025, η2p = 0.444). Lymphocytes, in NL with IBU were lower than in RL with IBU and PLA (*p* = 0.001, η2p = 0.491). Monocytes, in the NL with IBU and PLA, were lower than in the RL with IBU (*p* = 0.049, η2p = 0.344). For hemoglobin, hematocrit, and erythrocyte, the NL with IBU and PLA were higher than the RL with IBU and PLA (*p* < 0.05). Ammonia, with the use of IBU in the NL, obtained values higher than in the RL (*p* = 0.007), and with the use of PLA, the NL was higher than the RL (*p* = 0.038, η2p = 0.570). Conclusion: The training level tends to influence the immune system and, combined with the use of the IBU, it tends to improve recovery and the immune system.

## 1. Introduction

Paralympic powerlifting (PP) emphasizes muscle strength. Training is characterized by intensity, and recovery is an important factor for performance, where insufficient recovery can lead to losses in preparation and even injuries [1,2,3]. Furthermore, intense training tends to cause physiological stress with reflexes on immune responses and blood biomarkers [4], alterations which are reflected in the leukocyte count and the association of these leukocytes (neutrophils with monocytes/macrophages) [5,6].

In the same direction, strength training can inhibit the regeneration of muscle myofibers, the activation of satellite cells, and hypertrophy as a result of training [7,8,9]. There is no consensus on the best form of recovery [6,10,11], especially immersion in cold water and the use of non-steroidal drugs [12,13]. In this sense, the use of ibuprofen (IBU) has been practiced due to its ease of acquisition when compared to other methods and the good results it yields concerning performance in high-intensity activities [1,12]. Thus, the hypothesis presented was that the use of ibuprofen—in the post-training and recovery phase—tends to positively impact performance and immunity.

From the above, the objective was to analyze the use of ibuprofen on strength indicators (peak torque and fatigue index) and immunological blood biomarkers (leukocytes, neutrophils, erythrocytes, hemoglobin, hematocrit, CRP, ammonia, among others) in Paralympic Powerlifting at national and regional levels in the post-workout recovery period.

## 2. Materials and Methods

### 2.1. Study Design

Figure 1 shows the development of the research, and Figure 2 demonstrates the order and timing of the tests. The survey was developed by a familiarization (week 1) and the organization regarding the use of ibuprofen or placebo (IBU or PLA) was carried out in weeks 2 and 3 [14].

The use or nonuse of ibuprofen before training was randomly defined by drawing lots, with 50% for each condition.

The study design consisted of a familiarization (week 1) and recovery with or without the use of ibuprofen (weeks 2 and 3, with 50% IBU and 50% PLA in week 2, the condition being reversed in week 3). The regimen consisted of 400 mg of ibuprofen/placebo supplementation pre-ingestion: 15 min before training, training: total duration of three hours, collection: force measurements and blood markers, 1-RM: one repetition maximum.

The collections took place between 9 am and 1 pm. Assessments were made 30 min before the start of the intervention and immediately after the end of the training for force indicators and after that for blood indicators. The blood indicators were not collected before, considering that the blood was collected in the forearm, which could cause discomfort in training.

Before training, the athletes warmed up, starting with the upper limbs, using the following movements: (1) elbow extension, (2) shoulder rotation, and (3) shoulder abduction. In total, three sets of 10 to 20 repetitions were performed; The warm-up lasted 10 min [3,15]. Continuing, the athletes performed a specific warm-up, following what was directed in other studies by our team (30% of 1-RM, being initially three seconds in the eccentric phase x, one second in the concentric phase) and 10 faster repetitions (one second in the eccentric and concentric phases). During warm-up, verbal stimuli were given to the athletes [3,15].

The training consisted of five sets of five repetitions (5 × 5), with a load of 80% of 1-RM. The intervention training was composed of invariable resistance. In recovery, placebo pills made of wheat flour with sugar (PLA) and ibuprofen 400 mg (IBU) were used, where both cohorts ingested a tablet 15 min before training.

### 2.2. Sample

The group of paralympic athletes was composed of 20 males divided into two groups—national and regional levels. In the national level group (NL), four athletes had spinal cord injury below the eighth thoracic vertebra, two had poliomyelitis sequelae, two had lower limb malformations, and two had cerebral palsy. All athletes are nationally ranked with top 10 rankings in their respective bodyweight categories. The 10 regional-level (RL) athletes had the following disabilities: three had spinal cord injury below the eighth thoracic vertebra, one had poliomyelitis sequelae, three had lower limb malformations, two had an above-the-knee amputation, and one had cerebral palsy.

Exclusion criteria were: (1) not participating in any intervention and/or collection activity; (2) in the 24 h before collection, having participated in strenuous exercise; (3) consuming alcohol, caffeine, anti-inflammatories (including IBU), nutritional supplements (confirmed by interview); (4) having an allergy to ibuprofen; and (5) having muscle or joint damage, and/or reporting any changes in high blood pressure.

The sample size was defined based on a previous study [1], which demonstrated the effect size of partial squared eta (η2p) = 0.6 in the analysis of the influence of ibuprofen on neuromuscular aspects in Paralympic powerlifting (with the variable creatine kinase activity). The open-source G* Power (Version 3.0; Berlin, Germany) software was used in the statistical configuration of the “F” family tests (two-way ANOVA), and an α < 0.05 and a β = 0.80 were considered. Two conditions were adopted (PLA vs. IBU) in two moments (Before vs. After). From what was found, a minimum sample of six athletes, with a sample power of 0.80, was indicated for the present study.

Table 1 demonstrates the characterization of the sample.

### 2.3. Ethics

The sample consisted of athletes who voluntarily signed an informed consent form, under resolution 466/2012 of the National Research Ethics Commission—CONEP, of the National Health Council. The principles of the Declaration of Helsinki (1964, reformulated in 2013), of the World Medical Association, were followed. The study, a clinical trial, was registered (CAEE ID: 79909917.0.0000.55.46) and approved by the Ethics Committee in Research with Human Beings of the Federal University of Sergipe (UFS), under the technical statement number 2637882/2018.

### 2.4. Body Mass Analysis

An electronic wheelchair scale with a Micheletti platform (Micheletti^®^, São Paulo, Brazil), with a maximum capacity of 300 kg was used to determine the athletes’ body mass.

### 2.5. Maximum Training Load Analysis

The 1-repetition maximum (1RM) value was used to determine the training load. The test was incremental until reaching the maximum load to be lifted only once. In the situation of not being able to do a repetition, 2.4 to 2.5% of the test load was subtracted. The rest between attempts lasted 3–5 min [3,15].

The force was evaluated through the fatigue index (FI), the peak torque (PT), and a Chronojump force sensor (Chronojump^®^, BoscoSystem, Barcelona, Spain), with a capacity of 500 kg, output impedance of 350 ± 3 ohm, insulation resistance > 2000 cc, and input impedance 365 ± 5 ohm, and a 24-bit 80 Hz digital converter was used, fixed to the straight bench press with a steel chain. The software used was provided by the manufacturer. To calculate the torque, the distance between the sensor and the center of the joint was used [15,17].

The isometric peak torque (PT), fatigue index (FI), and rate of torque development (RTD) were determined. The PT was determined by the product of the isometric force peak between the force sensor cable attachment point and the adapted bench press bar. The PT (Nm) was obtained through the formula Nm = (M) × (C) × (H), where M = mass in kg, C = 9.80665, H = height of the bar up to the force sensor (0.45 m). An angle of 90° was adopted between the forearm and the upper arm, and this angle was confirmed by an angular amplitude measuring device, Model FL6010 (Sanny^®^, São Paulo, Brazil) (Figure 3).

The fatigue index (FI) was determined by the maximum contraction for a period of 5.0 s and was calculated by the equation: FI = {(PT Maximum—PT Minimum/PT Maximum) × 100}. The rate of torque development (RTD) (N/s), was determined using the peak torque-to-time ratio until reaching the maximum torque (RTD = Δpeak torque/Δtime), at 300 ms [15,18,19].

### 2.6. Blood Sample Collection, Blood Cells, Leukocytes Count (HEMOGRAM), and Ammonia

Blood was collected by two nursing professionals from the antecubital vein of the forearm. It was analyzed at the Hospital of the Federal University of Sergipe (Aracaju, Brazil), according to another study carried out by our group [1].

Blood cell counts were evaluated on a five-part differential hematology analyzer (Beckman Coulter AcT 5 diff AL Hematology Analyzer, Brea, CA, USA). The analyzer utilizes sequential dilution and dual focus flow fluid dynamics technology employing the Coulter Principle of impedance [1,12]. The red blood cell count (erythrocytes), hematocrit, hemoglobin level, and white blood cells count (leukocytes) together with the subpopulation percentage were taken for further evaluation.

Biochemical analyses of the blood samples were collected, and the C-Reactive Protein (CRP) was determined through the turbidimetric method (LABEST brand, model LabMax 240), adapted by Cayres et al. [20] and Roberts et al. [21].

To avoid the loss of volatile compounds, plasma ammonia was immediately measured. The ammonia concentration was measured using an enzymatic UV method at 340 nm spectrophotometer (BioespectroModel SP-22 UV/Visible, Minas Gerais- Brazil) and kit (Randox^®^, Crumlin, UK) adapted by Prado et al. [22].

### 2.7. Post-Workout Recovery Using a Placebo

The no ibuprofen, placebo (PLA) cohort received two wheat flour with sugar capsules, packaged identically to the IBU, 15 min before training. The same protocol used for the IBU was used, as described below.

### 2.8. Post-Workout Recovery Using Ibuprofen

The study followed the study protocol of our research group [1,23], which used ibuprofen (IBU) 15 min before training. IBU supplementation was given through two IBU capsules (400 mg), totaling 800 mg. The PLA and IBU were packaged in identical capsules, and the experiment was double-blind (i.e., athletes and evaluators did not know what was in the capsules). Athletes were instructed on capsule ingestion. The procedures were monitored to ensure the feasibility of the research.

### 2.9. Statistical Analysis

Measures of central tendency, mean (X) ± standard deviation (SD) were used. Considering the sample size, normality was determined using the Shapiro–Wilk test. To assess the differences between the data for the national and regional level groups, the independent Student’s “*t*” test was used. For the *t*-test, an effect size (Cohen’s “*d*”) was considered, adopting values of low effect (≤0.20), medium effect (0.20 to 0.80), high effect (0.80 to 1.20), and very high effect (>1.20) [24,25]. For the blood and force tests between the levels (national vs. regional), and the use or not of ibuprofen (PLA vs. IBU), the ANOVA (two-way) test was used. The Bonferroni test was used to assess point differences. In the evaluation of force indicators, MIF, FI, and RTD, an ANOVA was used for repeated measures in relation to the moments (before and after), supplement use (PLA and IBU), and training level (national and regional). In ANOVA, the effect size was evaluated through the “partial square eta” (η2p), adopting values of low effect (≤0.05), medium effect (0.05 to 0.25), high effect (0.25 to 0.50), and very high effect (>0.50) [24,25,26]. The program used was the Statistical Package for the Social Science, version 22.0 (IBM Corp., Armonk, NY, USA) considering that the significance level adopted was *p* < 0.05.

## 3. Results

Figure 4 presents data related to the peak torque fatigue index and rate of torque development in the use of placebo (PLA) and ibuprofen (IBU) at different training levels.

The results presented indicate for: Figure 4A peak torque (PT) “a” indicates a difference between before and after, with PLA, in NL; and “b” in RL (F = 3.051, η2p = 0.253, high effect). “c” indicates difference in the moment after, in NL between PLA and IBU (F = 24.959, η2p = 0.735, very high effect). In the fatigue index (FI). Figure 4B, “a, b, c, d” indicates a difference of NL and RL with PLA and IBU between the before and after moments (F = 2273.917, η2p = 0.996, very high effect). There was still “e” difference in the NL at the time after between the PLA and IBU (F = 4.253, η2p = 0.321, high effect), and “f” indicates difference with IBU between NL and RL (F = 119.288, η2p = 0.930, very high effect). Figure 4C, rate of torque development (RTD), “a” indicates a difference in NL with the use of IBU between the before and after moments (F = 5.545, η2p = 0.381, high effect). “b” indicates a difference in the before moment in NL between the use of PLA and IBU, the same occurring “c” in RL (F = 34.898, η2p = 0.795, very high effect). There was still a “d” difference in the NL at the after moment between the PLA and the IBU (F = 5.908, η2p = 0.396, high effect).

Table 2 and Figure 5, Figure 6, Figure 7 and Figure 8 present the data related to the blood markers in the use of placebo (PLA) and ibuprofen (IBU) at different training levels.

Regarding the results, the data presented point to: Figure 5A, leukocytes, “*” indicates a difference in ibuprofen (IBU) between regional and national levels (F = 4.405, η2p = 0.329, high effect). Figure 5B, neutrophils, “*” indicates a difference in IBU in national related to regional level with IBU and placebo (PLA) (F = 7.187, η2p = 0.444, high effect). Figure 5C, lymphocytes, “*” indicates the difference in IBU from national to regional level with IBU and PLA, (F = 8.675, η2p = 0.491, high effect). Figure 5D, monocytes, “*” indicates the difference in IBU and PLA in national related to regional level with IBU, (F = 4.712, η2p = 0.344, high effect).

Figure 6 presents the data related to the eosinophils and basophils in the use of placebo (PLA) and ibuprofen (IBU) at different training levels.

The results indicate: Figure 6A, for eosinophils, there were no differences. Figure 6B, for basophils, “*” indicates a difference in regional level PLA and IBU (F = 19.656, η2p = 0.686, very high effect), “**” in ibuprofen (IBU) and placebo (PLA) in the national level, and PLA in the regional level (F = 14.240, η2p = 0.613, very high effect).

Figure 7 presents the data related to ammonia as a biomarker of peripheral muscle fatigue with the use of placebo (PLA) and ibuprofen (IBU) at different training levels.

Figure 8 presents the data related to the hematological markers, (A) erythrocytes, (B) hemoglobin, (C) hematocrit, and (D) C-reactive protein (CRP) as an inflammation stage marker with the use of placebo (PLA) and ibuprofen (IBU) at different training levels.

Regarding the results, the findings indicate that: Figure 8A erythrocytes, “*” indicates a difference in ibuprofen (IBU) and placebo (PLA) at the regional and national levels (F = 6.761, η2p = 0.429, high effect). Figure 8B, hemoglobin, “*” indicates a difference in IBU and PLA in the regional and national levels (F = 8.998, η2p = 0.501, very high effect). Figure 8C hematocrit, “*” indicates a difference in IBU and PLA in the regional and national levels (F = 10.047, η2p = 0.527, very high effect). Figure 8D CRP did not indicate differences.

## 4. Discussion

The research aimed to evaluate the use of IBU before training in PP, at national and regional levels, regarding force and blood biochemical indicators. The results showed that regarding peak torque, with the use of IBU and PLA at the national level, there were differences, with no difference occurring at the regional level. In FI, there were differences at the national level between the IBU and PLA, and there were differences between the FI with the use of IBU between the national and regional levels.

The use of IBU improves peak torque in national-level athletes. Thus, there was less fatigue with the use of IBU for Paralympic powerlifting (PP) athletes when compared to PLA. A similar result was observed in another study [23] that showed a decrease in gastrocnemius muscle fatigue in runners who used IBU.

Regarding fatigue, the generation of force tends to decrease with fatigue and tends to impair movement’s motor control [27,28]. The origins of fatigue can be central or peripheral. In this direction, it can promote disorder in the movement sequence [29], and recovery can be impaired [30]. On the other hand, the possible change in joint kinematics may interfere with greater or lesser fatigue in strength training work [31]. However, any changes in supplementation [3,12] or even in training devices, with a change in the movement pattern, can cause increased fatigue [32]. The same occurred with magnesium supplementation on muscle soreness and performance. Magnesium supplementation reduced post-intervention muscle soreness at 24, 36, and 48 h, which did not happen with the placebo group. Recovery, muscle soreness, and perceived exertion were improved, and consequently fatigue, with the use of magnesium [33].

Concerning blood indicators linked to the immune system, changes in the number of leukocytes, neutrophils, lymphocytes and monocytes (immune parameters) were observed. Regarding lymphocytes, the findings indicate that there would not be time for the production of new cells in the bone marrow, which would promote the mobilization of the cellular matrix of these extra blood cells [6]. White blood cell count tends to increase, and associated with exercise, would induce an increase in circulating stress hormones. These hormones would mobilize white blood cells [34,35]. The increase in circulating leukocytes would be due to training, which would influence the production and release of the mentioned hormones [36]. In contrast, a lack of physical activity tends to reduce the body’s systemic antioxidant defense [37].

Concerning muscle damage, this induces the accumulation of neutrophils and cytokines, potentiating oxidative stress [38]. In this sense, exercise would improve the adaptive and antioxidant defense system [39,40]. On the other hand, the use of IBU would delay the inflammatory response after exercise, improving performance in weight training [1], justifying the decrease in fatigue indicated in the use of IBU in the study. High-intensity or exhaustive strength training would be related to increased oxygen consumption, promoting increased production of reactive oxygen species (ROSs) and an increased possibility of injuries and fatigue [41]. In turn, non-steroidal anti-inflammatory drugs (NSAIDs) have been used all over the world [42,43]. Among them, IBU has been presented as one of the most used. Its effect would be related to analgesic, antipyretic and anti-inflammatory effects used in the decrease of acute pain due to inflammatory processes. The ibuprofen used would be rapidly absorbed, and the dose of 400 mg had a peak concentration of 20–40 mg/mL in 1 to 2 h and returned to values of 5 mg/mL at the end of 6 h [44]. IBU would present rapid absorption, inhibiting cyclooxygenase. On hematocrit, platelet count was similar between IBU and PLA [45], which corroborates our findings, since there were no differences between IBU and PLA.

In the case of ammonia, in the PLA there was a significant difference both in passive recovery and the use of IBU, both at regional and national levels. Ammonia tends to be generated from different sources, being recognized as an important energy metabolite for exercise, especially resistance. Thus, during training, the increase in ammonia levels in the blood and brain tends to harm the central nervous system and lead to fatigue [46]. In contrast, maintaining blood glucose during training tends to decrease fatigue and aid performance [47]. Different from this, the level of urea would be related to that of proteins, dehydration, stress, and fatigue [48]. Thus, ammonia, since it is also involved in this process, is a by-product of the metabolism of nitrogenous compounds and is involved in several metabolic reactions [49]. This marker is indicative of liver dysfunction, commonly found in acute and chronic liver injury/failure, which can lead to disturbances in physical and cognitive-motor performance, and impairs memory, motor coordination, and decision making [50].

When evaluating neuromuscular fatigue after high-intensity resistance exercise (five sets with 80% of 1 RM) in athletes, it was found that resistance exercises with high loads tend to influence neuromuscular and biochemical processes in addition to fatigue parameters, where the levels of ammonia and lactate tend to be high [51]. Still, concerning ammonia, where the blood levels of ammonia and lactate obtained in real training conditions were evaluated, it was shown that the levels of ammonia in the blood and lactate obtained during repeated sprints were higher in male athletes. Another point that corroborates our study was that the maximum concentrations of ammonia in the blood were found between training sessions. Thus, ammonia in the blood would provide the trainer with additional information about the level of degradation of adenosine triphosphate and the contribution of the energy system involved, allowing greater control of training [52]. Ammonia as a training control factor and protein degradation has been suggested, and resistance training associated with chronic liver disease tends to beneficially affect muscle mass and strength, a fact observed by the reduction of ammonia in the blood. Thus, the increase in ammonia tends to represent greater protein degradation [53]. Still in this direction, muscle fatigue after exercise has been evaluated by several biochemical indicators, including ammonia levels [54].

In our study, the most trained athletes had a higher level of ammonia after training, indicating that they had a higher training intensity and therefore, protein degradation. However, recovery with the use of ibuprofen was satisfactory, demonstrating that it can be an important means to reduce fatigue and improve performance, even in more intense training.

Concerning physical exercise, the number of leukocytes, lymphocytes, neutrophils, and monocytes can be altered. A study evaluated two training regimens, high-intensity interval and moderate-intensity continuous training, on hematological biomarkers in men. Differences were observed in the count of leukocytes (*p* < 0.01), lymphocytes (*p* < 0.05), neutrophils (*p* < 0.05), and monocytes (*p* < 0.01), with an increase for continuous and a decrease for activities of high intensity. Thus, high-intensity exercises such as strength training would not be as effective concerning the immune response [55]. Another study emphasizes that hypertrophy would be dependent on eccentric muscle actions and the resulting inflammatory response, which was not addressed in our study. In this study, untrained (UTG) and trained (TG) people were evaluated. The muscle damage levels were higher at 2 and 24 h after training for the untrained and 24 h for the trained. Neutrophils increased in both groups immediately after and 2 h after exercise. Lymphocytes increased after exercise but decreased 2 h after training in both groups, while monocyte numbers increased only immediately after the intervention. This protocol altered the count/total number of circulating immune cells in both groups [56].

Ibuprofen is a non-steroidal anti-inflammatory drug [57], which tends to decrease hormonal activity linked to pain and inflammation. Ibuprofen is normally used to reduce fever and the inflammatory process, as well as the pain of various etiologies. Thus, the excessive use of ibuprofen would be associated with liver damage, among others [58]. Regarding the use of ibuprofen before a cycling event, it evaluated the response of circulating leukocytes. Leukocyte subsets increased. Thus, it appears that the consumption of ibuprofen tends to attenuate anti-inflammatory cytokine IL-10 increases but would not alter leukocyte subset concentrations [59]. Perhaps an explanation for this is that Paralympic athletes have a greater amount of body musculature involved in strength training, such as the bench press, as mentioned in another study regarding hemodynamic variation in conventional and Paralympic powerlifting athletes, which would explain part of the findings in our study [60].

Regarding temperature, the inflammatory process is characterized by an increase in local temperature. Thus, skin temperature tends to increase during high-intensity anaerobic exercise, decreases afterwards, and tends to increase again in the following days after exercise [61]. In this sense, the use of ibuprofen could mask or hinder this trend in terms of local temperature. Contrary to that, the local temperature, measured through thermography, showed no correlation with blood CK levels, pain level, perception of recovery, and perception of fatigue (r < 0.2, *p* > 0.05) [62]. In the same direction, when approaching athletes with a spinal cord injury (SCI), difficulties in maintaining thermal balance during training were demonstrated due to lower sweating capacity and cutaneous vasodilation. However, the results were inconclusive, making it difficult to determine a parameter for this follow-up like some of our athletes [63].

However, the use of ibuprofen did not show good results related to the immune system, where the ingestion of drinks with carbohydrates during intense and prolonged exercises attenuated the increase in plasma cytokines and stress hormones, superior to the use of ibuprofen, however, partially limited [64]. The use of ibuprofen alone or in combination with other drugs in patients affected by Duchenne muscular dystrophy, being evaluated in maximum concentration, did not affect these indicators [65]. Yet another study that evaluated the influence of ibuprofen use on hypertrophy showed that ibuprofen presented a reduction in inflammation-induced muscle weakness. The use of ibuprofen combined with exercise presented a protective effect against age-related muscle loss, thus increasing muscle mass and performance. However, the mechanism related to muscle strength and hypertrophy seems to differ between the elderly and young, but the effects would be more dose-related and would be related to training capacity [66].

What was found in our study is in agreement with several other studies. On the other hand, other studies should be carried out to determine which type of physical activity would have the greatest influence on the immune system.

The limitations of our study would be the lack of control over the athletes’ diet as well as the lack of control over the subjects’ sleep quality. The exposition on psychological stress factors in the living environment ought to be considered and measured. These points, such as food, stress, and sleep control, can have some effect on the immune system or even interfere with recovery.

## 5. Conclusions

The use of ibuprofen had a positive effect on isometric strength, fatigue, and rate of torque development, which did not occur with the placebo. There were no differences in blood indicators related to the immune system, except for basophils, indicating that this indicator was higher in regional athletes who used the placebo. The other indicators point out that the level of training tends to influence the indicators related to the immune system, being, together with the use of IBU, an important ally in post-workout recovery.

The practical applications of the study indicate that the use of ibuprofen would be an effective form of recovery after training or even targeting a specific competition, and in addition to recovery, the use of ibuprofen tends to impact the immune system at the moment after training. From the above, it was also observed that more experienced athletes at the national level tend to be able to implement greater intensity in training, a fact that was verified through the concentration of ammonia in the blood. However, the recovery tends to be better, and the use of ibuprofen tends to corroborate this recovery.

We emphasize that ibuprofen is a medication and, when used in an uncontrolled way, can lead to side effects such as kidney damage, dyspepsia (stomach burning), nausea, heartburn, dizziness, blurred vision, ringing in the ears, fluid retention, edema, constipation, excess gas, itching, and decreased urinary volume, which are normally linked to frequent use. Our study used ibuprofen acutely in a single opportunity and should not be recommended as a long-term therapy.

## Figures and Tables

**Figure 1 healthcare-10-01331-f001:**
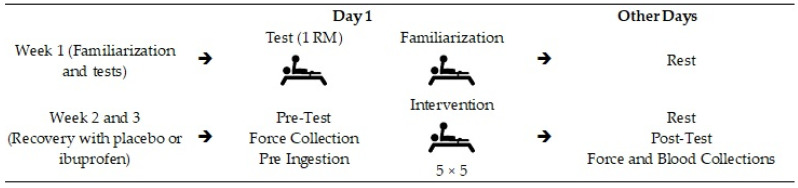
Study design. Legend: → Indicate the order in which the procedures were performed.

**Figure 2 healthcare-10-01331-f002:**
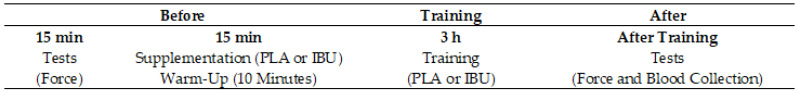
Intervention timeline.

**Figure 3 healthcare-10-01331-f003:**
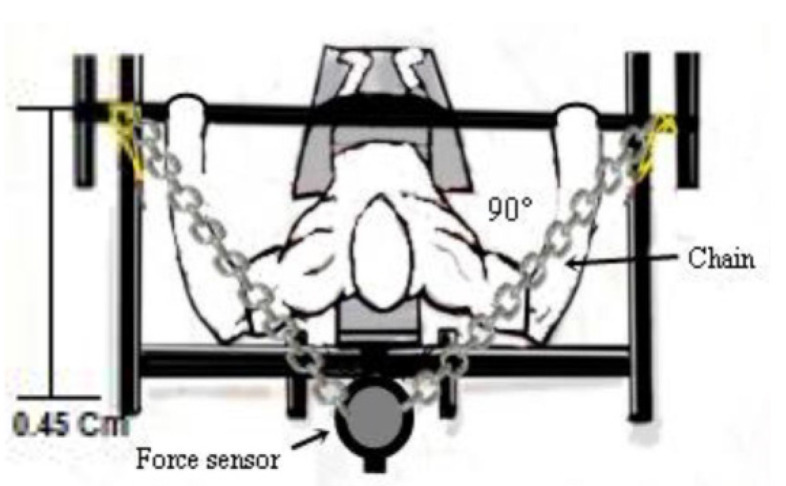
Demonstration of the force sensor attachment.

**Figure 4 healthcare-10-01331-f004:**
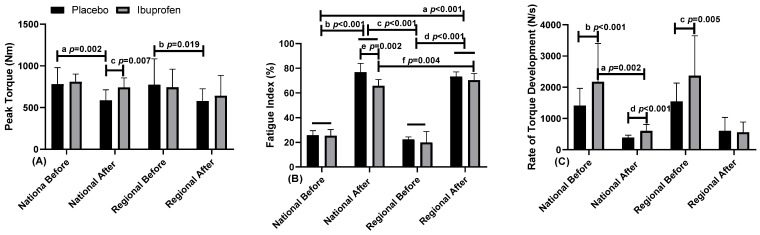
(**A**) Peak torque; (**B**) fatigue index; (**C**) rate of torque development in the use of placebo (PLA) and ibuprofen (IBU) at the regional and national levels of training. Legend: * (Letters) *p* < 0.05.

**Figure 5 healthcare-10-01331-f005:**
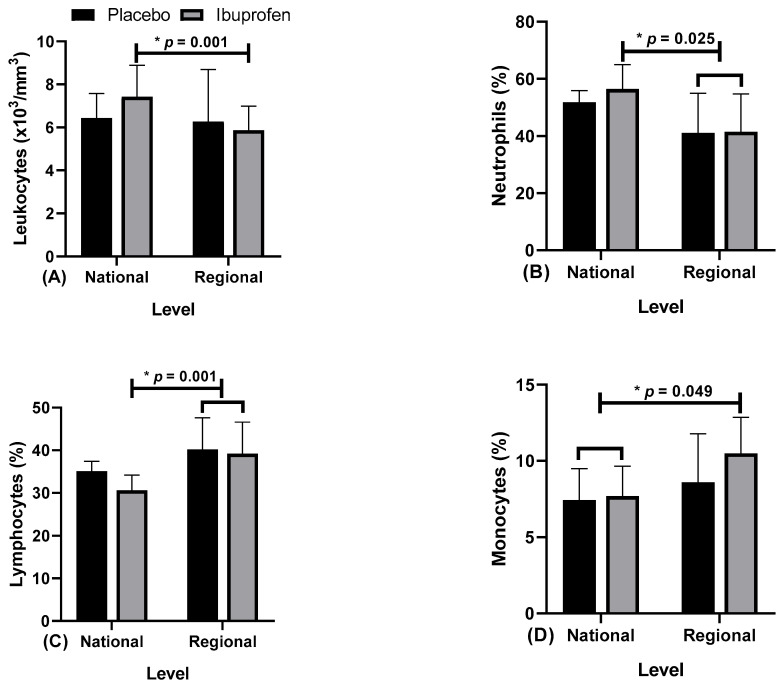
Blood markers: (**A**) leukocytes; (**B**) neutrophils; (**C**) lymphocytes; and (**D**) monocytes; with the use of placebo (PLA) and ibuprofen (IBU) at the regional and national levels of training. * *p* < 0.005.

**Figure 6 healthcare-10-01331-f006:**
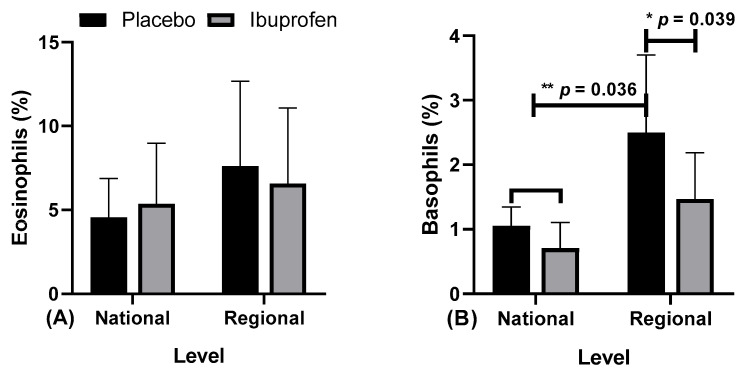
Blood markers: (**A**) eosinophils and (**B**) basophils in the use of placebo (PLA) and ibuprofen (IBU) at regional and national levels of training. * and ** *p* < 0.05.

**Figure 7 healthcare-10-01331-f007:**
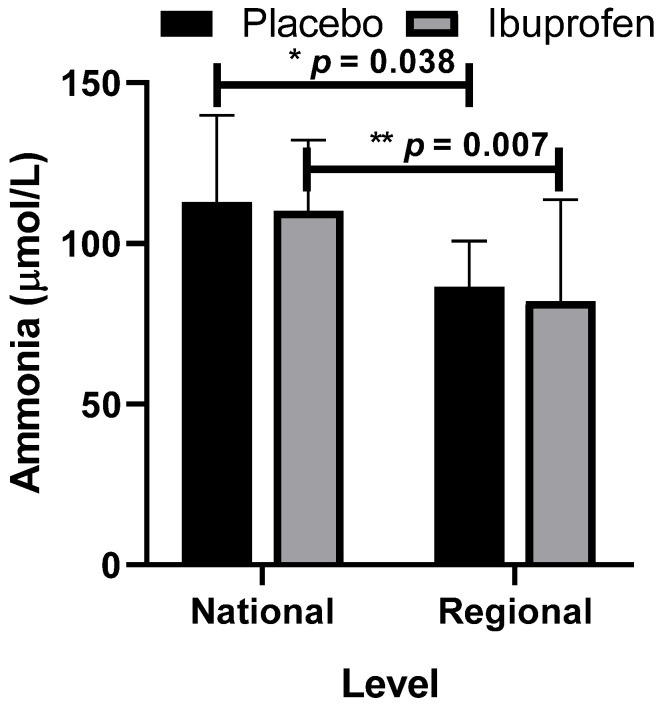
Ammonia with the use of placebo (PLA) and ibuprofen (IBU) at different training levels. Regarding the results, “*” indicates a difference between the regional and national levels with the use of placebo (*p* = 0.038) and “**” with the use of ibuprofen (*p* = 0.007, F = 11.937, η2p = 0.570, very high effect).

**Figure 8 healthcare-10-01331-f008:**
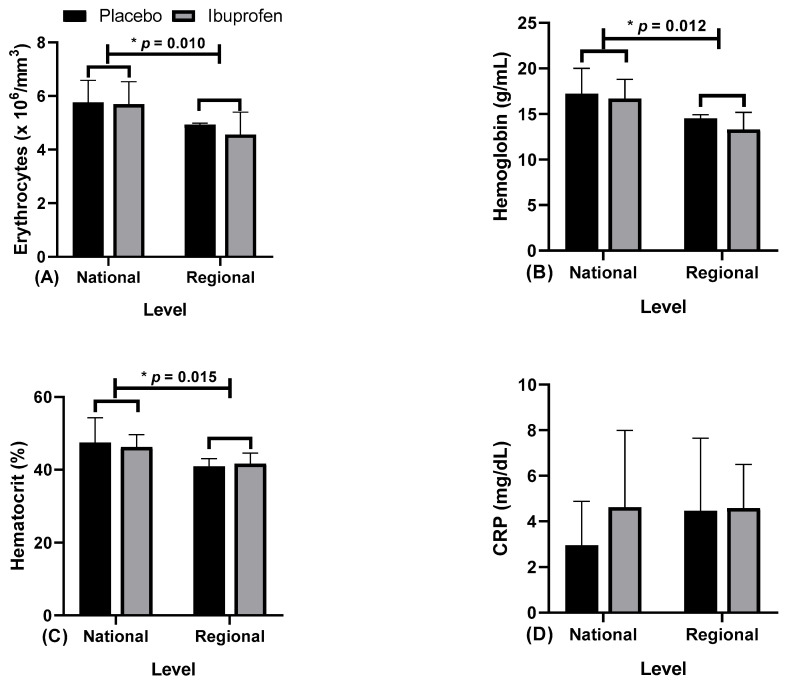
Blood markers: (**A**) erythrocytes; (**B**) hemoglobin; (**C**) hematocrit; and (**D**) C-reactive protein (CRP) with the use of placebo (PLA) and ibuprofen (IBU) at the regional and national levels of training. * *p* < 0.005.

**Table 1 healthcare-10-01331-t001:** Sample characterization.

Variables	National Level (X ± SD)	Regional Level (X ± SD)	*p*	*t*	Cohen’s *d*
*n*	10	10			
Age (years)	32.50 ± 3.00	30.75 ± 5.32	0.055	0.975	0.427
Body mass (kg)	84.00 ± 17.63	74.50 ± 33.88	0.305	0.963	0.352
Experience (years)	3.80 ± 0.68	3.18 ± 0.24	0.068	0.913	0.216
1-RM/Bench press (kg)	153.75 ± 20.56 *	123.00 ± 33.46	0.046 #	2.930	1.061a
1-RM/Body Weight	1.86 ± 0.21 **	1.77 ± 0.41 **	0.038 #	1.419	0.419b

# *p* ≤ 0.05 (Independent “*t*” test); “a” high effect (0.80 a 1.20); “b” medium effect (0.20 to 0.80). * All athletes ranked in the top 10 in their bodyweight categories at the national level. ** They had an index greater than 1.4 in the bench press (1-RM/Body Weight), being considered elite, according to Ball and Wedman [16].

**Table 2 healthcare-10-01331-t002:** Blood markers with the use of placebo (PLA) and ibuprofen (IBU) at different training levels.

	National Level (NL)		Regional Level (RL)		ANOVA Effects		
	PLA	IBU	PLA	IBU	Supplem	Level	Interact
Leukocytes (×10^3^/mm^3^)	6.44 ± 1.15	7.42 ± 1.47 *	6.27 ± 2.43	5.87 ± 1.128 *	F = 0.350 η2p = 0.037 *p* = 0.569	F = 4.405 η2p = 0.329b *p* = 0.001 *	F = 1.708 η2p = 0.160 *p* = 0.224
Neutrophils (%)	51.84 ± 4.08	56.46 ± 8.52 *	41.07 ± 13.91 *	41.50 ± 13.27 *	F = 1.038 η2p = 0.103 *p* = 0.335	F = 7.187 η2p = 0.444b *p* = 0.025 *	F = 0.394 η2p = 0.042 *p* = 0.546
Lymphocytes (%)	35.12 ± 2.27	30.60 ± 3.58 *	40.20 ± 7.39 *	39.20 ± 7.41 *	F = 2.816 η2p = 0.128 *p* = 0.238	F = 8.675 η2p = 0.491b *p* = 0.001 *	F = 1.000 η2p = 0.100 *p* = 0.343
Monocytes (%)	7.42 ± 2.08 *	7.68 ± 1.98 *	8.59 ± 3.19	10.49 ± 2.37 *	F = 2.853 η2p = 0.241 *p* = 0.125	F = 4.712 η2p = 0.344b *p* = 0.049 *	F = 0.954 η2p = 0.096 *p* = 0.354
Eosinophils (%)	4.56 ± 2.31	5.36 ± 3.63	7.62 ± 5.05	6.57 ± 4.52	F = 0.036 η2p = 0.004 *p* = 0.853	F = 0.997 η2p = 0.100 *p* = 0.344	F = 1.174 η2p = 0.115 *p* = 0.307
Basophils (%)	1.05 ± 0.29 *	0.71 ± 0.39 *	2.50 ± 1.21 *#	1.47 ± 0.72 #	F = 14.240 η2p = 0.613a *p* = 0.036 *	F = 19.659 η2p = 0.686a *p* = 0.039 #	F = 1.627 η2p = 0.153 *p* = 0.234
Erythrocytes (×10^3^/mm^3^)	5.77 ± 0.82 *	5.70 ± 0.84 *	4.93 ± 0.05 *	4.57 ± 0.83 *	F = 2.436 η2p = 0.213 *p* = 0.153	F = 6.761 η2p = 0.429b *p* = 0.010 *	F = 1.282 η2p = 0.125 *p* = 0.287
Hemoglobin (g/mL)	17.22 ± 2.80 *	16.70 ± 2.09 *	14.50 ± 0.43 *	13.30 ± 1.87 *	F = 3.280 η2p = 0.267 *p* = 0.104	F = 8.998 η2p = 0.501a *p* = 0.012 *	F = 3.197 η2p = 0.262 *p* = 0.107
Hematocrit (%)	47.46 ± 6.84	46.21 ± 3.49	40.93 ± 2.07	41.69 ± 2.90	F = 0.039 η2p = 0.004 *p* = 0.849	F = 10.047 η2p = 0.527a *p* = 0.015 *	F = 1.161 η2p = 0.114 *p* = 0.309
C-Reactive Protein (mg/dL)	2.96 ± 1.92	4.62 ± 3.36	4.46 ± 3.20	4.58 ± 1.92	F = 5.885 η2p = 0.195 *p* = 0.058	F = 0.594 η2p = 0.062 *p* = 0.461	F = 1.236 η2p = 0.121 *p* = 0.295
Ammonia (μmol/L)	112.90 ± 27.04 *	110.23 ± 21.97 *	86.57 ± 14.26 #	82.05 ± 31.64 #	F = 0.132 η2p = 0.014 *p* = 0.725	F = 11.937 η2p = 0.570a *p* = 0.038 * *p* = 0.007 #	F = 0.029 η2p = 0.003 *p* = 0.869

* or # *p* ≤ 0.05 (ANOVA two way and post hoc de Bonferroni); * NL; # RL; “b” high effect (0.25 to 0.50); and “a” very high effect (>0.50); PLA: placebo; IBU: ibuprofen; Supplem: Supplement; Intercat: Interation.

## Data Availability

The data that support this study can be obtained from the address: www.ufs.br/Department of Physical Education, accessed on 12 December 2021.

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
