# Peer review of "Evaluation of Ibuprofen Use on the Immune System Indicators and Force in Disabled Paralympic Powerlifters of Different Sport Levels"

_healthcare, 2022, doi:10.3390/healthcare10071331_

Round 1
Reviewer 1 Report
This study was to analyze whether the IBU would affect performance and immune system after training in Paralympic Powerlifting (PP). The authors indicated that the level of training tends to influence the immune system and combined with the use of the IBU tends to improve recovery and the immune system.
1. The purpose of the present study was to evaluate the different levels of training combined with Ibuprofen on the immune system and powerlifting performance, there are a couple of issues that need to be concerned.
(a) The authors revealed the effects of Ibuprofen and placebo on the blood markers of national and regional athletes. Moreover, one of the clinical effects of Ibuprofen is decreasing body temperature, which is related to inflammation regulation. Regarding exercise training may alter the numbers of leukocytes, lymphocytes, neutrophils, and monocytes, for a better understanding that the effects of Ibuprofen on exercise-induced- immune response, the baseline of blood markers before training, and the changes in body temperature before and after training could be provide and discuss.
(b) Regarding exercise training-induced immune response and related to skeletal muscle regeneration and hypertrophy, which is able to impact exercise performance. Does ibuprofen intake suppress the benefits of skeletal muscle regeneration and hypertrophy evoked by the exercise-induced immune response?
2. The content of “placebo” needs to be clarified. On line 113, the authors indicated that Placebo was used through the use of wheat flour; on line 193, the authors indicated that Placebo condition received two wheat flour with sugar capsules.
3.Some editing for English language is required throughout the manuscript.
Author Response
Dear Reviewer,
Please see our answers.
Best regards,

Reviewer 2 Report
This is a very interesting manuscript addressing a challenging question on appropriate misconceptions in the context of Ibuprofen and post-workout recovery. Further, a faster recovery for athletes is a topic of recent interest. Despite well written and showing interesting data interpretation, there are some problematic points. I have disagreements about their conclusions. The authors do not have sufficient data to make inferences regarding “immune system”. A leucogram is very superficial for claiming immune system. Although there are many institutions involved in this work, there are few analyses evaluating physiological characteristics and complications in Powerlifters using Ibuprofen. I have comments and questions that I have detailed below.
- The age of groups (national and regional level) is different and could disturb interpretations. The age should be matched.
- In the abstract: The training consisted of 5x5 (five sets of five repetitions) on the bench press. In this phrase must be described the intensity (%RM or load in kilograms).
- This phrase in the abstract can be improved: The use of IBU provided a higher PT in National Level, (p=0.007, ɳ2p=0.347), a lower FI in National IBU compared PLA (p=0.002, ɳ2p=0.635) and National IBU compared Regional (p=0.004, ɳ2p=0.414). The last part (National IBU compared Regional) leads to a misinterpretation.
- In the abstract: Leukocytes, Neutrophils and Lymphocytes showed differences between National and Regional with IBU (p=0.001, ɳ2p=0.329). It is necessary to describe more precisely what exactly is meant under (showed differences). Increased or reduced?
- In the abstract: For hemoglobin, hematocrit and erythrocyte, there were differences between national and regional (p=0.012, ɳ2p=0.501). It is necessary to describe more precisely what exactly is meant under (there were differences). Increased or reduced?
- This phrase in the introduction can be reduced (very big sentence): This, coupled with the fact that there is disagreement regarding the best forms of recovery [6,10,11], which ends up bringing the possibility of several forms of recovery or even methods that can help or induce an increased recovery have used immersion in water cold, and even the use of non-steroidal drugs for recovery, without however having a consensus on the methods [12,13].
- The authors of paper are right in augmenting recovery strategies, but I suggest that the use of Ibuprofen (IBU) be more emphasized in the introduction. It must be described pharmacological properties of Ibuprofen (a nonsteroidal anti-inflammatory drug). The inclusion of more literature can strengthen the manuscript This may better explain why post-training recovery should be increased athletes using Ibuprofen.
- Regarding the context of manuscript, I detected that Erythrocytes, Hemoglobin, and Hematocrit data are not strongly related to Immune System. I believe that these data should be less emphasized (only in text form, but not in figures).
- I recommend more detailing in ANOVA results. The F-and P values must be shown for the effect of Ibuprofen (PLA x IBU), effect of level (National × Regional) and interactions. These results could be described in text or within the figure.
- “On the other hand” is excessively used over the manuscript.
- About the aims of study “From the above, the objective was to analyze the use of ibuprofen on strength indicators and on immunological blood biomarkers in Paralympic Powerlifting at national and regional level in the post-workout recovery period”. I suggest the inclusion of variables related to strength indicators and on immunological blood biomarkers.
- In the phrase: The isometric torque peak (PT) was determined by the product of the isometric force peak, between the force sensor cable attachment point and the adapted bench press. The authors should consider the inclusion of picture showing the adaptation related to force sensor cable attachment.

Author Response

(The authors gave the same response as above.)

Round 2
Reviewer 1 Report
This study was to determine whether Ibuprofen would affect performance and immune system after training in Paralympic Powerlifting. The results revealed that the level of training tends to influence the immune system and combined with the use of the Ibuprofen tends to improve recovery and the immune system.
1. Line 106-107, the authors revealed that “Assessments were made 30 minutes before the start of the intervention and immediately after the end of the training”. The results of the current manuscript only showed the data of “after”. For more understanding of the effect of Ibuprofen on performance and the immune response, the results of “before” ” should be also provided.
2. Figure legend of Figure 5, please provide the description of “**”.
3. The manuscript needs English language editing.
Author Response
Dear Reviewer,
Please see the responses to comments.
Best regards,

Reviewer 2 Report
With regard the manuscript “Evaluation of Ibuprofen use on the immune system indicators and strength in disable powerlifters of different sport level”. Despite showing improvements in manuscript, I still have disagreements about the care in detailing methods and results. The quality (precision of information) need to be improved.
· With regard the question (The age of groups is different and could disturb interpretations. The age should be matched), the answer of authors was that athletes at the national level already train longer and would normally start in the sport around 20 years old. However, in table 1, the experience is around 3.8 (years). Something looks strange..
· Figure 1: The word Familiariztion is wrong.
- Lines 165-179 can be improved (the method is very disorganized).
- I recommend more detailing in Chronojump force sensor. Explanations about load cell records, signals capture, conversion and signals treatment are lacking.
- The authors report that height of the bar in relation to the load cell (0.45 m), but it is not clear if this height (vertical distance) was used in calculus for determining torque. For obtaining an adequate measure of Torque (Force x Distance), I understand that should be adopted the distance between sensor and the hand of athlete (force application point). “The center of the joint was used” which joint are you referring ?
- Line 177: were obtained through the formula Nm = (M) × (C) × (H), where M = Body mass in Kg, C = 9.80665. Why body mass? Would not be the mass of bar and plates?
- The authors add a good picture showing the adaptation related to force sensor cable attachment. However, this figure may be used for detailing all distances involved in experiment. In the figure 2, I suggest the inclusion of hypothetical example using numbers (demonstrating the mathematical rationale for obtaining force and torque).
· With regard the results, only peak torque is showed, but you have force data that could be analyzed (Remembering that torque is force x distance). The authors should report more analyses related to force and torque in average values. Moreover, if there was a monitoring of force for a period of 5 seconds, it would be interesting to apply the trapezoidal method (calculating the area under the curve) to determine the force over the entire 5-s period (impulse). This is important to readers for didactical reasons (better understand your protocol).
- Line 117: The training consisted of five sets of five repetitions (5x5), with a load of 80-90% of 1-RM. Why 80-90% of 1-RM. Just an intensity (80%) should be used to maintain the study consistency.
· Caution with their conclusion. The authors do not have sufficient data to make inferences regarding “immune system”.

Author Response

(The authors gave the same response as above.)

Round 3
Reviewer 2 Report
With regard the manuscript “Evaluation of Ibuprofen use on the immune system indicators and strength in disable powerlifters of different sport level”. Still, there are some issues to be addressed. I perceive a neglect in detailing of information of manuscript (disposition of authors is lacking):
· With regard the table 1, comparisons (by using t test) between National and Regional Level must be described.
· I still have some doubts with regard the study design. For example, how much training combined with supplementation were, in fact, done. A more elucidative figure should be added. Below is an example according to my interpretation of text.
· Lines 175-177> The isometric torque peak (PT), Fatigue Index (FI) and Rate of Torque Development (RTD), was determined by the product of the isometric force peak, between the force sensor cable attachment point and the adapted bench press. This phrase must be well written.
- Explanations about mathematical rationale for obtaining force and torque must be well written (especially in lines 182-183).
- In Statistical Analysis (lines 221-229): Now authors inserted more results in Figure 3. Was used ANOVA (for repeated measures) to compare changes in strength outcomes before and after (within-individual variation)? Statistical procedures must be revised.
- I insist, authors must provide all statistical information’s. For each variable, the F-and P values must be shown for each main effect: supplementation effect (F=… , P=..), level effect (F=… , P=..), time effect (F=… , P=..) and well as their interactions (F=… , P=..). The insertion of a table containing all ANOVA findings and post hoc comparisons would be better for didactic purposes. Below is an example:
|
National level |
|
Regional level |
|
ANOVA effects |
||||
|
Placebo |
Ibuprofen |
|
Placebo |
Ibuprofen |
|
Supplementation |
Level |
Interactions |
Erythrocytes |
X±SD |
X±SD |
|
X±SD |
X±SDab |
|
F=xxx, ɳ2p=xxx, P=xxx |
F=xxx, ɳ2p=xxx, P=xxx |
F=xxx, ɳ2p=xxx, P=xxx |
Hemoglobin |
X±SD |
X±SD |
|
X±SDab |
X±SD |
|
F=xxx, ɳ2p=xxx, P=xxx |
F=xxx, ɳ2p=xxx, P=xxx |
F=xxx, ɳ2p=xxx, P=xxx |
Hematocrit |
X±SD |
X±SD |
|
X±SD |
X±SD |
|
F=xxx, ɳ2p=xxx, P=xxx |
F=xxx, ɳ2p=xxx, P=xxx |
F=xxx, ɳ2p=xxx, P=xxx |
Leukocytes |
|
|
|
|
|
|
….. |
|
|
Post hoc differences: a, b, c, d indicate significant differences (p < 0.05) in relation to Placebo-NAT, Ibuprofen-NAT, Placebo-REG and Ibuprofen-REG, respectively.
- Caption of figure 3 must contain more information about time between before vs after.
· With regard results, it is confused to identify differences between groups (for example figure 3B when comparing placebo vs Ibuprofen in national before*p<0.001). This seems strange given that averages look similar.
· Conclusion in lines (438-443): The use of ibuprofen had a positive effect on isometric force, fatigue and rate of torque development, which did not occur with placebo. This showed differences in indicators related to the immune system, in relation to the use of IBU when compared to the use of PLA. What “indicators related to the immune system” are you referring? Only Basophils showed differences between IBU vs PLA (within the same level). The effects of ibuprofen are dependent of national and regional level. The authors must be more specific and prudent when writing their conclusion.

Author Response
Dear Reviewer,
Please see the responses.
Thank you!

This manuscript is a resubmission of an earlier submission. The following is a list of the peer review reports and author responses from that submission.